# How Are Bystanders Involved in Cyberbullying? A Latent Class Analysis of the Cyberbystander and Their Characteristics in Different Intervention Stages

**DOI:** 10.3390/ijerph192316083

**Published:** 2022-12-01

**Authors:** Yanru Jia, Yuntena Wu, Tonglin Jin, Lu Zhang

**Affiliations:** School of Psychology, Inner Mongolia Normal University, Hohhot 010000, China

**Keywords:** bystanders of cyberbullying, bystander intervention, college students, latent class analysis

## Abstract

Background: Cyberbullying is a phenomenon that occurs by means of digital devices in virtual environments. Although research reveals the relevant role played by bystanders in stopping cyberbullying, the patterns of cyberbullying bystanders among Chinese college students is not clear. Data: Participants were 1025 Chinese college students (62.0% girls, 38.0% boys). The present analyses empirically explored the roles of cyberbystanders (passive outsider online, defender of the cybervictim online, reinforcer of the cyberbully online, passive face-to-face outsider, face-to-face defender of the cybervictim, and face-to-face reinforcer of the cyberbully) using latent class analysis. Results: (1) Five latent classes were identified: defensive bystander (17.9%), indifferent bystander (10.1%), low-involved bystander (10.2%), medium-involved bystander (45.7%), and high-involved bystander (16.0%). (2) The cyberbystander patterns varied significantly for all stages of bullying intervention, among which the defensive bystander had the lowest score in the notice stage but the highest scores in the other stages. (3) There was a graded relationship between the five latent classes and the level of social network site use and cyber-victimization experience. College students with high usage of social network sites and high cyber-victimization experience were more likely to engage in diverse bystander behaviors.

## 1. Introduction

Bystanders of cyberbullying refers to “individuals who witness cyberbullying”, and these bystanders can be divided into three types, according to their behavioral responses after witnessing cyberbullying: defender, reinforcer, and outsider [1]. The defender plays a vital role in protecting others from cyberbullying, and the bystander shelter behavior can quickly and effectively curb a bully’s behavior and alleviate harm to those being bullied [2], which impacts psychological factors, such as moral cognition, high empathy, and self-efficacy [3,4]. In contrast, by giving the bully positive feedback (e.g., encouragement or laughter), the reinforcer strengthens the bully’s aggressive behavior against the victim, enlarging the situation and causing secondary damage [5]. Outsiders refer to cyberbystanders who stand on the sidelines when someone is being bullied, or who focus only on protecting themselves. Studies have shown that 59 to 70% of college students have witnessed cyberbullying on social media. However, few people will react positively or negatively to it; most people choose to remain an outsider and allow the cyberbullying to continue [6,7]. In addition to the above three common behavioral responses, Sarmiento et al. [8] further considered the environmental factors, believing that bystanders would exist in face-to-face interactions (such as talking to or sitting with people involved in cyberbullying beside him) and completely online (such as interacting on social networking sites). That is divided into six types passive outsider online, defender of the cybervictim online, reinforcer of the cyberbully online, passive face-to-face outsider, face-to-face defender of the cybervictim, and face-to-face reinforcer of the cyberbully. Considering the large number of bystanders to cyberbullying and the impact of their various behaviors on forming healthy social environments, it is important to further explore the patterns cyberbullying bystanders in the context of college students. The analysis of the bystander model of cyberbullying has been divided into three parts: latent class analysis, intervention stage characteristics, and behavior-influencing factors.

### 1.1. Latent Classes of Bystanders in Cyberbullying

There are significant differences in the behavioral responses of the different types of cyberbystanders, however previous studies have focused primarily on the diversity of the bystanders’ roles, and analyzed from the single perspective of the defender, reinforcer, or outsider [9]. However, this framing makes it difficult to explore the internal heterogeneity of the cyberbystander population and examine the cross-population. In the case of cyberbullying, the same bystander may play different roles simultaneously, such as someone assisting in cyberbullying (i.e., the reinforcer) and also acting as a witness to the cyberbullying (i.e., the outsider) [10]. Is there then an intersection between the complex and diverse roles of cyberbystanders? What kind of behavioral commonalities might this cross-group have? A latent class analysis (LPA) can potentially help us answer these questions. 

A LPA is an individual-centered statistical method that can explain the correlation between explicit continuous variables through potential continuous variables, fully using all of the sample data to estimate the possibility of which group all individuals belong to. The basic assumption is that the probability distribution of various responses of explicit variables can be explained by a small number of mutually exclusive potential category variables, and the choice of explicit variables in each category has a specific tendency [11]. A LPA can make up for the disadvantages of variable-based measurements, allowing the exploration of in-group classifications and helping us understand which individuals show unique behavior patterns, in order to provide more targeted interventions for the various subgroups and improve the number of protections. Furthermore, it can also effectively integrate different information, compare subgroups more intuitively and comprehensively, and explore cross-group characteristics. Therefore, the current study used a potential profile analysis to explore the heterogeneity of cyberbullying bystanders in both network and face-to-face situations.

### 1.2. Intervention Characteristics of Cyberbystanders

Bystanders are essential to cyberbullying events [12]. Their intervention can alleviate or aggravate the victim’s bullying dilemma [13]. However, little is known about when and why bystanders intervene. The bystander intervention model (BIM) can help us explain the conditions under which people choose to help or not help people being bullied [14]. Five stages have been identified as the steps that must take place for a bystander to intervene. Specifically, one must (a) notice the event, (b) interpret the event as an emergency that requires help, (c) accept responsibility for intervening, (d) know how to intervene or provide help, and (e) implement intervention decisions. Studies have found that 68 percent of people notice when cyberbullying occurs, but only 10 percent directly intervene [15], meaning that there are many missing links interrupting the action from the cyberbystander. So, where is the point where the steps are disrupted, preventing the bystander from becoming an active protector? This study used these five aforementioned stages of bullying intervention (i.e., notice, emergency, responsibility, know, act) to investigate the behavioral response characteristics of the different types of cyberbystanders in these stages. Furthermore, the study aimed to uncover the hindering stages in the process of bullying intervention.

### 1.3. Influencing Factors of Cyberbystanders

According to the ecological systems theory, individuals are embedded in a series of environmental systems that influence one another. The micro-system is closely related to individual behavior [16]. To further clarify the predictive factors leading to the formation of the subcategory of cyberbystanders, the current study aimed to explore cyberbystanders from two aspects: individuals in the micro-system (perceived anonymity and cyber-victimization experience) and environment (social network use). First, the study chose perceived anonymity as the predictor of different bystander subcategories, starting from individual factors. Perceived anonymity refers to one’s belief in their ability to maintain an unrecognizable status in a specific digital environment [17]. Due to the online disinhibition effect, perceived anonymity can reduce the cost and adverse consequences of online behavior, leading individuals to engage in more unconstrained online behavior. For example, Lapidot-Lefler and Barak [18] found that anonymous participants were more likely to use threatening language in online debates than those who were not anonymous. Therefore, we can reasonably assume that the higher one’s perceived anonymity, the more likely one is to engage in involved bystander behavior (protective or reinforcement behavior). Second, the cyber-victimization experience is another significant individual predictor. Studies have found that cyber-victimization may simultaneously lead to both more helpful and hurtful behaviors. For example, Barlinska et al. [19] found that being bullied in an online environment is a factor that increases the negative behavior of bystanders. Meanwhile, DeSmet et al. [20] suggested that those who had previously been victims of cyberbullying showed a stronger intention to help other victims. As a result, the cyber-victimization experience can affect the potential bystander category, both online or face-to-face. Finally, social networking site use is an environmental factor that affects the behavior of cyberbystanders. Routine activity theory suggests that the digital monitoring ability and online lifestyle of individuals—such as frequent click on online icons without precaution—increases their likelihood of becoming victims of cyberbullying [21]. Therefore, based on prior research and ecological systems theory, we explored the predictive effects of perceived anonymity, cyber-victimization experience, and social network site use on the subcategories of college students cyberbystanders.

To examine the data, we used latent class analysis to explore the cyberbystander patterns of college students, focusing on the intervention characteristics and influencing factors of different model subgroups. The current study proposed the following hypotheses: (1) potential subgroups of cyberbystanders can be observed; (2) there are differences in the performance of different potential categories of bystanders in different stages of cyber-bystander intervention; and (3) subgroups of cyberbystanders are associated with perceived anonymity, cyber-victimization experience, and social network site use.

## 2. Materials and Methods

### 2.1. Participants and Procedure

A total of 1025 college students were selected using a random cluster sampling from four Chinese cities, which were Shandong Province, Henan Province, Fujian Province and Inner Mongolia Autonomous Region (male = 389; female = 636). The participants were representative of the overall sample in terms of age, gender, grade, and place of origin (i.e., whether the participants came from rural or urban areas). The participants ranged from 17 to 23 years of age (*M*_age_ = 20.05, *SD* = 1.23); were in their first to fourth year of study, with a breakdown of 202 (19.7%), 339 (33.1%), 287 (28.0%), and 197 (18.2%); and included 318 students (31.0%) in rural areas, 707 (69.0%) students in urban areas, respectively. The study obtained the informed consent of all participants, as well as the approval of the local ethics committee before commencing.

### 2.2. Measures

Cyberbullying bystander scale (CBS). The 40-item scale was used to evaluate different groups of bystanders in cyberbullying [8]. The instrument is based on the idea that all cyberbullying bystanders could exist in two situations: face-to-face interactions (e.g., talking to or sitting next to a person involved in cyberbullying) and online (e.g., interacting on a social networking site). As such, the CBS is divided into six dimensional structures: passive outsider online, defender of the cybervictim online, reinforcer of the cyberbully online, passive face-to-face outsider, face-to-face defender of the cybervictim, and face-to-face reinforcer of the cyberbully. Participants rate their responses on a 5-point Likert scale ranging from 1 (never) to 5 (very frequently). Cronbach’s α for the scale in the current study was 0.92.

Bystander intervention in bullying scale (BIBS) [22]. The 16-item scale consists of five dimensions (i.e., notice, emergency, responsibility, know and act), which measure the corresponding stages of the bystander intervention model. Higher scores represent a higher degree of the bystander’s awareness and behavior of interfering with bullying. Participants rate their responses using a 5-point Likert scale ranging from 1 (strongly disagree) to 5 (strongly agree). Cronbach’s α for the scale in the current study was 0.83.

Anonymity and strength differential scale (ASDS) [23]. The 10-item ASDS consists of two dimensions that describe one’s agreement with Anonymity and Strength attitudes. The Anonymity subscale comprises five items (e.g., “I feel comfortable sending mean text messages or e-mails to anybody regardless of whether I know them or not.”), all of which are rated on a 5-point scale ranging from 1 (strongly disagree) to 5 (strongly agree). Higher scores represent a higher degree of perceived anonymity. Cronbach’s α for the scale in the current study was 0.88.

Social network site intensity scale (SNSI) [24]. The 8-item SNSI consists of two dimensions that measure the extent to which the participant is actively engaged in social networking site activities (e.g., “Social networking sites are part of my everyday activities.”). Higher scores represent a higher frequency of use intensity of social networking sites. Cronbach’s α for the scale in the current study was 0.78.

Cyberbullying inventory (CBI) [25]. The 18-item CBI consists of one dimension that describes one’s frequency of cyber-victimization (e.g., “I’ve been hurt by people I know online”). Higher scores represent a higher frequency of experiencing cyber-victimization. Participants rate their responses using a 4-point Likert scale ranging from 1 (have never encountered) to 4 (encountered more than five times). Cronbach’s α for the scale in the current study was 0.96.

### 2.3. Data Analysis

First, we screened and calculated the descriptive statistics for all measurements. Second, to test for the existence of the discrete groups (classes) with similar psychometric profiles, we conducted a LCA using the dichotomous items of the CBS. Then, on this basis, we analyze the intervention characteristics of different cyberbystanders in the above stages (MANCOVA). Finally, multivariate logistic regression models were used to examine the associations of the CBS and perceived anonymity, cyber-victimization experience, and intensity of social network sites.

## 3. Results

### 3.1. Identification and Delimitation of the Latent Classes of Cyberbullying Bystanders

In this study, using the LCA of the 40 items of the CBS, six potential category models were established, and their fitting indexes were compared and selected for classification. Table 1 shows the fit indices resulting from the different estimated LC models. With the increase in the number of categories, the values of AIC, BIC, and aBIC continued to decrease. The entropy classification index was close to 1, indicating that the model gradually improved. The six-class model did not replicate the best log likelihood values and the LMR was not statistically significant (*p* > 0.05), so it was therefore not considered further. The five-class model was the best fitting solution, when compared to the one-, two-, three-, and four-class models. The five-class model presented a statistically significant LMR *p*-value and the lowest AIC and BIC values. 

Membership for the five-class solution was as follows (see Figure 1). Latent class 1 (LC1, *n* = 183, 17.9%) includes individuals with high scores on item 6 to 11 (defender of the cybervictim online) and items 24 to 32 (face-to-face defender of the cybervictim) and was labelled “defensive bystander”. Latent class 2 (LC2, *n* = 103, 10.1%) consisted of individuals with high scores on items 1 to 5 (passive outsider online), medium scores on items 6 to 11 (defender of the cybervictim online) and items 19 to 32 (passive face-to-face outsider, face-to-face defender of the cybervictim). Considering the obvious passive bystander characteristics of class 2, we called it “indifferent bystander”. Lastly, Latent Class 3 (LC3, *n* = 104, 10.2%), Latent Class 4 (LC4, *n* = 468, 45.7%), and Latent Class 5 (LC5, *n* = 164, 16.0%) consisted primarily of individuals with balanced scores on the CBS items and, thus, were called “low-involved bystander”, “medium-involved bystander”, and “high-involved bystander”, respectively. 

### 3.2. Characteristics of the Cyberbullying Bystanders in Different Stages of the Bullying Intervention

A MANCOVA revealed that the cyberbystander patterns varied significantly for all stages of a bullying intervention. Table 2 shows the mean scores, standard deviations, and effect sizes resulting from the analyses. A clear differentiation was found that showed that defensive bystanders had the lowest score in the notice stage, but the highest score in all other stages. High-involved bystanders scored higher in all five stages.

### 3.3. Multiple Logistic Regression Analysis

A multinomial logistic regression was used to assess the association between the influencing factors and the latent class of the CBS. For details, influencing factors included anonymity, SNS intensity, cyber-victimization experience, and participants’ sociodemographic variables, such as gender (female vs. male), grade (1th–4th grade), and left-behind pattern (categories: “not left-behind”, “father migrant”, “mother migrant”, and “two-parent-migrant”).

Table 3 indicated that, compared to the low-involved bystander class, one group was seen to be positively related to the SNS intensity, with OR (95% CI) of 1.91 (1.22~2.97) for the high-involved bystander class (*p* < 0.05). Similarly, two groups were seen to be positively related to the cyber-victimization experience, with OR (95% CI) of 3.52 (2.16~5.74) and 3.99 (2.33~6.84) for the medium- and high-involved bystander classes (*p* < 0.01). 

## 4. Discussion

### 4.1. Co-Occurrence and Heterogeneity of the Cyberbullying Bystander Subcategory

First, the results of the LCA showed that there were five potential patterns of cyberbullying bystanders among college students: defensive bystander (17.9%), indifferent bystander (10.1%), low-involved bystander (10.2%), medium-involved bystander (45.7%), and high-involved bystanders (16.0%). Overall, there were significant differences in the scores and trends of the different roles of cyberbullying bystanders, indicating the heterogeneity in this group. Among the five classes, there were four bystander groups with a high involvement, with the low-involved bystander class being the exception. The total proportion of the four groups was 89.8%, which shows that cyberbullying bystanders are common among college students in China.

Second, the score of each cyberbullying bystander subcategory also had a multi-role co-occurrence. That is to say that individuals who demonstrate a certain kind of bystander behavior will simultaneously exhibit other forms of bystander behavior as well. For example, the trends of six bystander behaviors for medium- and high-involved bystanders are similar, and the level of each is relatively consistent. In addition to the three equilibrium-involved classes of bystanders, this study found two unique classes of defensive and indifferent bystanders with unique co-occurrence phenomena. Among them, the defensive bystanders scored higher in the two types of online and face-to-face defense behaviors, while outsiders and reinforcers scored lower. The results showed that college students in this category tend to stop and interfere when noticing cyberbullying, whether communicating with the bullies face-to-face or online. Indifferent bystanders scored higher in the passive outsider online category, and medium in terms of online and face-to-face defender categories. The results indicated that this category of college students responds less to events in the face of cyberbullying. The network environment further reduces their intervention intention, making their inaction more prominent online.

### 4.2. Intervention Characteristics of the Cyberbystanders

The analysis of variance showed significant differences in the five stages of bystander intervention among college students in the different subcategories of the cyberbullying bystander. For defensive bystanders, among the five stages of bullying intervention, the score of the notice stage was significantly lower than that of other groups. However, the scores for the other four stages of emergency, responsibility, know, and act were higher. It means that although defensive bystanders may not be acutely aware of cyberbullying, it is easy for them to enact positive and protective interventions once they discover it happening. For high-involved bystanders, bullying intervention scored higher in all five stages, which was also the highest scoring overall with the exception of defensive bystanders. This shows that high-involved bystanders are drawn quickly into cyberbullying. Whether it is a positive or negative behavior, they will participate in it in their own way and may become affected by the emotions or atmosphere. In addition, the reason that prevents low-involved bystanders and indifferent bystanders from making positive interventions may be the group members’ low sense of responsibility and intervention ability. For them, even if they have noticed the occurrence of the incident, the scattered responsibilities in the cyber incident and the huge onlookers may reduce their intention to intervene in the cyberbullying incident. Meanwhile, the low intervention behaviors of the largest number of the low-involved bystanders (45.7%) are more likely, due to their negative interpretation of events, which aligns with the findings of existing studies. For example, one study found that most participants believe that being bullied is the fault of the victim, and that bystanders believing the victim is responsible for their plight will significantly reduce the chances that the bystanders will provide social support to the victim [26,27].

### 4.3. Influencing Factors of Cyberbystanders

The results showed that the experience of cyber-victimization and the intensity of social networking sites could predict the cyberbullying bystanders category group to which some college students belong. Specifically, the more experience one has of cyber-victimization, the more likely they will be a medium- or high-involved bystander, that is, the experience of cyber-victimization will affect the bystanders’ degree of involvement of cyberbullying. This result was in line with the social reconnection hypothesis which states that when individuals experience damage to social ties, they will often try to repair it through subsidiary behavior [28]. People will obey authority [29] and even spend money to regain social connections [30], which shows the power of the motivation to reintegrate into society [31]. Therefore, whether it is active protection or malicious enhanced bullying, individuals may increase their connections with the group in various ways in order to reintegrate themselves. Meanwhile, the higher the intensity of one’s use of social networking sites, the more likely they will become a high-involved bystander. This is in line with the daily activity theory of the Internet, which suggests that the higher the intensity of one’s use of social networking sites, the more likely they are to witness cyberbullying.

Furthermore, the analysis of the different models’ demographic characteristics showed differences between the demographic variables, such as gender and left-behind model. For gender, there is no gender difference in the defensive bystanders, which is consistent with previous studies [32]. However, on the whole, in comparison to girls, boys are more likely to be involved as cyberbystanders, and this involvement includes both indifference and positive intervention. The results support the role continuity theory that boys are more likely to show role overlap than girls in cyberbullying. As a result, it may be because boys’ awareness of the seriousness of cyberbullying is significantly lower than that of girls, so they are more likely to participate in the event in the form of “joyous”. In addition, boys who prefer highly active online activities, such as online games, are more likely to be involved in cyberbullying than girls [33]. Regarding the left-behind patterns, compared with those not left-behind, it appears to be easier to predict the involvement of bystanders who were left behind by their mother alone or by both parents, than those who were left behind by their father alone. In other words, those with a migrant father were less likely to be indifferent and medium-involved bystanders. At the same time, a migrant mother predicted defensive and high-involved bystanders, which may be related to the parents’ roles in the student’s life to this point. According to the object relation theory, when object relevance between an individual and their father or mother is destroyed, and external situations force the child to be separated from their father, mother, or both parents, this may induce the individual to demonstrate more degenerative bad behavior [34].

## 5. Limitation

There are still some limitations in this study, which need to be valued and improved. On the one hand, although the sample selected was statistically representative, it was only from four cities in China. Future research should include individuals from different Chinese regions. On the other hand, due to the features of the cross-sectional data, it is difficult to explain the causal relationship between variables and observe the cyberbullying bystanders’ dynamic changes in the time domain. Future studies can further explain the causality and strengthen the horizontal and vertical comparative studies of potential profiles to understand better this group’s characteristics and behavior patterns, that is, how one profile is transformed into another and what factors led to this transformation.

## 6. Conclusions

Overall, this study provides additional evidence for the heterogeneity of college students’ cyberbullying bystander behavior in the context of China. In this study, through the investigation of face-to-face and online CBS, we found five CBS subgroups related to the intensity of social networking site use and the experience of cyber-victimization, and the degree strength of correlation was different. Therefore, the results suggest that when formulating intervention plans to improve the behavior of active cyberbystanders, we should focus on a different CBS for a targeted intervention.

## Figures and Tables

**Figure 1 ijerph-19-16083-f001:**
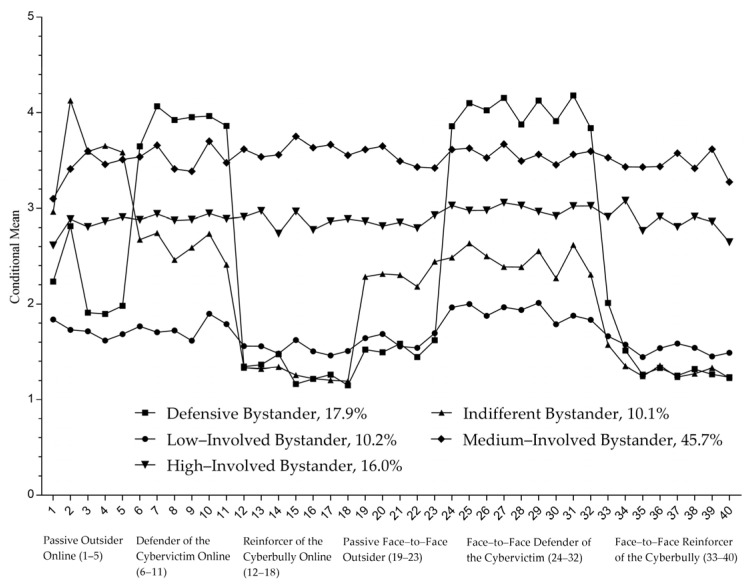
Five classes of the cyberbullying bystander (CBS) of the best fitting five class pattern.

**Table 1 ijerph-19-16083-t001:** Fit indices for the latent class models on the cyberbullying bystander for the total sample.

Model	K	Log	AIC	BIC	aBIC	Entropy	LMR	BLRT	Class Probability
1	80	−64,614.53	129,389.05	129,783.65	129,529.56				
2	121	−59,669.95	119,581.89	120,178.72	119,794.41	0.98	0.00	0.00	0.40/0.60
3	162	−57,713,50	115,751.00	116,550.06	116,035.53	0.98	0.00	0.00	0.20/0.20/0.60
4	203	−56,938.43	114,282.86	115,284.15	114,639.40	0.95	0.00	0.00	0.19/0.46/0.20/0.16
5	244	−56,360.86	113,209.71	114,413.23	113,638.26	0.95	0.00	0.00	0.10/0.10/0.18/0.46/0.16
6	285	−56,003.19	112,576.39	113,982.13	113,076.95	0.96	0.12	0.00	0.09/0.07/0.14/0.09/0.46/0.16

Note. AIC = Akaike information criterion; BIC = Bayesian information criterion; aBIC = adjusted BIC; LMR = the Lo–Mendell–Rubin adjusted likelihood ratio test; BLRT = bootstrapped likelihood ratio test.

**Table 2 ijerph-19-16083-t002:** Mean scores on bullying intervention as a function of the resulting latent classes.

	Defensive Bystander	Indifferent Bystander	Low-Involved Bystander	Medium-Involved Bystander	High-Involved Bystander	*F*	η^2^*_p_*
Notice	2.65 ± 1.00	2.82 ± 0.95	2.93 ± 0.97	2.76 ± 0.79	3.14 ± 1.00	8.17 **	0.03
Emergency	3.57 ± 0.98	3.31 ± 0.92	3.40 ± 0.89	3.01 ± 0.80	3.43 ± 0.85	12.53 **	0.05
Responsibility	3.42 ± 1.05	3.29 ± 0.93	3.11 ± 0.88	3.03 ± 0.81	3.42 ± 0.89	10.71 **	0.04
Know	3.46 ± 0.95	3.37 ± 0.80	3.11 ± 0.82	3.11 ± 0.77	3.41 ± 0.89	8.82 **	0.03
Act	3.35 ± 1.86	3.16 ± 0.92	2.98 ± 0.89	2.91 ± 0.81	3.23 ± 0.95	11.67 **	0.04

Note. ** *p* < 0.01.

**Table 3 ijerph-19-16083-t003:** Association of the influencing factors with different latent classes of the CBS.

Factors		Defensive Bystander		Indifferent Bystander		Medium-Involved Bystander		High-Involved Bystander	
		OR	CI (95%)	OR	CI (95%)	OR	CI (95%)	OR	CI (95%)
Gender	Female	1.00		1.00		1.00		1.00	
	Male	1.25	0.71~2.19	1.95 *	1.06~3.56	2.01	1.22~3.31	2.19 **	1.27~3.76
Grade	Grade 4	1.00		1.00		1.00		1.00	
	Grade 1	1.03	0.47~2.26	1.38	0.59~3.26	1.94	0.98~3.83	1.84	0.87~3.87
	Grade 2	1.47	0.74~2.90	1.53	0.71~3.26	2.15	1.17~3.97	1.53	0.77~3.03
	Grade 3	1.57	0.80~3.10	1.12	0.51~2.44	1.45	0.78~2.71	1.26	0.63~2.52
Left-behind pattern	Not Left-Behind	1.00		1.00		1.00		1.00	
	Father Migrant	0.41 *	0.19~0.88	0.26 **	0.10~0.71	1.22	0.66~2.26	1.44	0.73~2.82
	Mother Migrant	0.23 *	0.07~0.83	0.38	0.11~1.37	2.97 *	1.30~6.80	2.88 *	1.20~6.93
	Two-Parent-Migrant	0.58	0.30~1.13	0.59	0.28~1.23	0.56	0.30~1.03	0.56	0.27~1.13
Anonymity		0.95	0.66~1.36	0.84	0.57~1.21	0.85	0.62~1.15	0.75	0.52~1.09
SNS Intensity		1.40	0.92~2.13	1.18	0.75~1.87	0.81	0.56~1.18	1.91 *	1.22~2.97
Cyber- Victimization Experience		1.02	0.58~1.78	0.85	0.45~1.60	3.52 **	2.16~5.74	3.99 **	2.33~6.84

Note. * *p* < 0.05, ** *p* < 0.01.

## Data Availability

Due to privacy issues, the data may not be shared publicly.

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
