# Peer review of "How Are Bystanders Involved in Cyberbullying? A Latent Class Analysis of the Cyberbystander and Their Characteristics in Different Intervention Stages"

_ijerph, 2022, doi:10.3390/ijerph192316083_

Round 1
Reviewer 1 Report
Thank you for the opportunity to review “How are bystanders involved in cyberbullying? A latent class analysis of the cyberbystander and their characteristics in different intervention stages” in the International Journal of Environmental Research and Public Health. The current study conducted latent class analysis (LCA) to examine the patterns of cyberbullying bystanders among Chinese college students, which makes up for the lack of existing research and has high theoretical and social value. I appreciate the authors’ analysis method used in the study and I have several comments below that the authors may consider.
1. In the Abstract section: "There was a graded relationship between the five latent classes and level of social network site use, victim experience, and anonymity". The Abstract is inconsistent with the Text of the paper.
2. This study used a latent class analysis of six bystander subtypes, while only three types were introduced in the Introduction section. Please add the relevant information in the Introduction section.
3. In this study, the college students are used as participants, and “...sample in terms of age, gender, grade, and place of origin (i.e., whether the participants came from rural or urban areas)”. It is necessary to add the demographic information about the place of origin.
4. In the Table 2, please mark the significant results.
5. What do you mean "rational bystander"? Why do you name it that? Please explain in detail the reasons for the naming of the group "rational bystanders".
6. Please add the limitations in the Discussion section. Maybe the author can provide several limitations that should be considered when interpreting the results.
Author Response
Dear Reviewer,
Please see the revisions in the attachment.
Sincerely
Jia Yanru

Reviewer 2 Report
This research is meaningful. However, there are several things that need to be clarified or added so that the information presented in the manuscript becomes more comprehensive, clear, and makes the reader not misunderstood.
1. Study 2 needs to be supplemented.
1) I speculate that the intervention stage was divided by the second scale/questionnaire, but what specific dimensions and how to divide were not listed. Please add them in the assessment tool section.
2) At each stage of the intervention, the LPA classification of cyberbullying bystanders should be listed, and then ANOVA will be performed to make the results are shown coherently without being abrupt.
2. In the abstract information section regarding participants, it says "the patterns of cyberbullying bystanders among Chinese adolescents..." This should be stated as “Chinese college students”, not “adolescents”.
3. Line 214,"contradictory-protective bystanders". This class is not a category component of this paper. Please revise it.
4. p.2: line 71: What do you mean "cyberbullying bystanders in both network and face-to-face situations"? Please further elaborate on the relevant content.
5. The scale used in the study shall be introduced in a relatively consistent format. Line 162 “The eight-item CBI”.
6. In "Multiple Logistic Regression Analysis...", please add the process description such as statistical methods.
7. Please unify the writing format of numbers. "p<.01" or "p<0.01".
Author Response

(The authors gave the same response as above.)
